# Characterization of *Parageobacillus* Bacteriophage vB_PtoS_NIIg3.2—A Representative of a New Genus within Thermophilic Siphoviruses

**DOI:** 10.3390/ijms241813980

**Published:** 2023-09-12

**Authors:** Eugenijus Šimoliūnas, Monika Šimoliūnienė, Gintarė Laskevičiūtė, Kotryna Kvederavičiūtė, Martynas Skapas, Algirdas Kaupinis, Mindaugas Valius, Rolandas Meškys, Nomeda Kuisienė

**Affiliations:** 1Department of Molecular Microbiology and Biotechnology, Institute of Biochemistry, Life Sciences Center, Vilnius University, Saulėtekio av. 7, LT-10257 Vilnius, Lithuania; monika.simoliuniene@gmc.vu.lt (M.Š.); gintare.laskeviciute@gmail.com (G.L.); rolandas.meskys@bchi.vu.lt (R.M.); 2Department of Microbiology and Biotechnology, Institute of Bioscience, Life Sciences Center, Vilnius University, Saulėtekio av. 7, LT-10257 Vilnius, Lithuania; nomeda.kuisiene@gf.vu.lt; 3Department of Biological DNA Modification, Institute of Biotechnology, Life Sciences Center, Vilnius University, Saulėtekio av. 7, LT-10257 Vilnius, Lithuania; kotryna.kvederaviciute@mif.vu.lt; 4Department of Characterisation of Materials Structure, Center for Physical Sciences and Technology, Sauletekio av. 3, LT-10257 Vilnius, Lithuania; martynas.skapas@ftmc.lt; 5Proteomics Centre, Institute of Biochemistry, Life Sciences Center, Vilnius University, Saulėtekio av. 7, LT-10257 Vilnius, Lithuania; algirdas.kaupinis@gf.vu.lt (A.K.); mindaugas.valius@bchi.vu.lt (M.V.)

**Keywords:** bacteriophage, siphovirus, NIIg3.2, thermophiles, *Parageobacillus*, genomic analysis

## Abstract

A high temperature-adapted bacteriophage, vB_PtoS_NIIg3.2 (NIIg3.2), was isolated in Lithuania from compost heaps using *Parageobacillus toebii* strain NIIg-3 as a host for phage propagation. Furthermore, NIIg3.2 was active against four strains of *Geobacillus thermodenitrificans*, and it infected the host cells from 50 to 80 °C. Transmission electron microscopy analysis revealed siphovirus morphology characterized by an isometric head (~59 nm in diameter) and a noncontractile tail (~226 nm in length). The double-stranded DNA genome of NIIg3.2 (38,970 bp) contained 71 probable protein-encoding genes and no genes for tRNA. In total, 29 NIIg3.2 ORFs were given a putative functional annotation, including those coding for the proteins responsible for DNA packaging, virion structure/morphogenesis, phage–host interactions, lysis/lysogeny, replication/regulation, and nucleotide metabolism. Based on comparative phylogenetic and bioinformatic analysis, NIIg3.2 cannot be assigned to any genus currently recognized by ICTV and potentially represents a new one within siphoviruses. The results of this study not only extend our knowledge about poorly explored thermophilic bacteriophages but also provide new insights for further investigation and understanding the evolution of *Bacilllus*-group bacteria-infecting viruses.

## 1. Introduction

*Parageobacillus* is a genus of Gram-positive, obligately thermophilic, spore-forming, chemo-organotrophic bacteria belonging to the family *Bacillaceae* [1,2]. Members of this genus are phylogenetically the most closely related to *Geobacillus* spp. and have been isolated from a variety of environments, ranging from local soils [3] or hay compost [4] to high-temperature surroundings such as geothermal soil [5] or hot springs [6]. Unsurprisingly, as genera derived from extreme environments, *Parageobacillus*, and its closest relatives *Geobacillus*, serve as sources of proteins that remain stable at high temperatures and are functional under extreme conditions. Consequently, these bacteria hold great potential in various biotechnological, industrial, and medical applications [7,8,9]. On the other hand, bacteria of these genera cause food spoilage or form biofilms on different abiotic surfaces, which generates significant financial losses in the food industry [10,11]. Thus, bacteriophages and phage-derived proteins, which have been demonstrated to act as natural antimicrobial agents [12,13], could be considered as alternative tools against detrimental bacteria of the aforementioned genera.

Endolysins encoded by bacteriophages are of a great interest because of their efficiency in controlling bacterial infections, preventing biofilm formation or eliminating food-borne pathogens [14,15]. Furthermore, endolysins from viruses-infecting *Thermus*, *Meiothermus*, and *Geobacillus* have shown high (thermo)stability and unusually broad lytic activity against Gram-negative and Gram-positive bacteria, making them targets for commercialization as antimicrobials [16]. However, only a limited number of functionally known, thermostable endolysins have been isolated from *Geobacillus* and/or *Parageobacillus* spp. infecting bacteriophages [8,17,18,19,20,21] to date. Additionally, the role of bacteriophages affecting thermophilic *Bacillus*-group bacteria remains under-investigated.

To date, only one bacteriophage with a completely sequenced genome, namely JGon-2020a (CP063417), annotated as a *Parageobacillus* phage, has been deposited in Genbank (accessed in NCBI database on 25 July 2023). However, neither the paper characterizing this phage nor the predicted open reading frames (ORFs) encoding for proteins or tRNAs have been annotated in the publicly available genome of this phage (accessed in NCBI database on 25 July 2023). On the other hand, two thermophilic bacteriophages, *Geobacillus* phages GVE3 [22], and E1 (GVE1) [23], have been demonstrated to infect the *Geobacillus thermoglucosidasius* strain DSM 2542^T^ and the *Geobacillus toebii* strain E26323 (DQ225186), respectively. Thus, given that *G. thermoglucosidasius* and *G. toebii* have been reclassified as *Parageobacillus thermoglucosidasius* and *Parageobacillus toebii*, respectively [2], phages GVE3 and GVE1 could be annotated as *Parageobacillus* phages. Furthermore, it remains unclear whether *Geobacillus* virus E2 (GVE2) [24] could be named as a *Parageobacillus* phage. The GVE2 was purified from its host *Geobacillus* sp. E263 (China General Microbiological Culture Collection Center accession no. CGMCC1.7046), but due to the lack of publicly available information, it is unclear whether *Geobacillus* sp. E263 is the same *G. toebii* strain E26323 or not. As of now, to our knowledge, no more phages that are active on *Parageobacillus* spp. have been characterized.

In this study, the biological characteristics and complete genome analysis of bacteriophage vB_PtoS_NIIg3.2 (referred here by its shorter name, NIIg3.2), were presented. The phage was isolated from the compost heaps at Vilnius University Botanical Garden located in Vilnius, Lithuania. The phage host range determination experiments revealed that NIIg3.2 was active not only on *P. toebii* strain NIIg-3, which was used for phage isolation, but also on four strains of *G. thermodenitrificans*. The phage showed a high-temperature plating profile and demonstrated an ability to form plaques from 50 to 80 °C. The phylogenetic analysis indicated that NIIg3.2 was distant from already known viruses and likely represented a new genus within the siphophages. Thus, the data presented here not only provide information on the morphology, physiology, and genetic diversity of *Parageobacillus* and *Geobacillus*-infecting virus but also offer new insights about virus–host interactions in dynamic ecosystems such as compost heaps.

## 2. Results

### 2.1. Host Range, Morphology, and Physiological Characteristics of the Phage

Bacteriophage NIIg3.2 was isolated from the compost heaps using the enrichment of phages in the source material technique, as described previously [25], with the local isolate *Parageobacillus toebii* strain NIIg-3 as a host. In further experiments, 46 phylogenetically closely related thermophilic bacterial strains (41 local isolates collected from the same compost heaps and five *Bacillus*-group spp. from DSMZ collection) were used to explore the host range of phage NIIg3.2 (Appendix A). It was demonstrated that *G. thermodenitrificans* strains PT-4, NIIg-2, PK-11, and PK-3, as well as *P. toebii* strain NIIg-3, were sensitive to NIIg3.2. The other tested *Geobacillus* spp. and *Parageobacillus* spp., as well as all of the tested strains of *Aeribacillus*, *Brevibacillus*, *Peribacillus*, and *Ureibacillus* spp., were found to be resistant to NIIg3.2.

Transmission electron microscopy (TEM) observations of phage NIIg3.2 virions (Figure 1) revealed particles that corresponded to the B1 morphotype in Bradley’s classification [26,27]. Based on the morphological characteristics, NIIg3.2 was a siphovirus characterized by an isometric head (62.08 ± 3.96 nm (*n* = 25) in diameter) and a noncontractile tail (218.37 ± 12.53 nm [*n* = 25] in length, and 10.44 ± 1.03 (*n* = 25) in width). Although neither the baseplate nor the tail fibers were clearly visible via TEM, tail hub, and tail fiber encoding ORFs, accordingly, ORF16 and ORF17 were identified during bioinformatics analysis and/or by proteomics methods (Appendix A).

To determine the optimal conditions for phage propagation, the effect of temperature on the efficiency of plating (e.o.p.) was examined. It was determined that phage NIIg3.2 infected host cells in the temperature range from 50 to 80 °C. The phage formed plaques with a clear center (up to 0.6 mm in diameter) surrounded by an opaque halo zone (up to 2 mm in diameter) (Figure 1B), indicating the presence of phage-encoded bacterial exopolysaccharide (EPS)-degrading depolymerases [28]. As all the tested thermophilic host strains demonstrated limited growth in liquid LB medium, it was not possible to perform the adsorption tests and/or single-step experiments under the investigated conditions.

### 2.2. Overview of Genome

Phage NIIg3.2 had a double stranded DNA genome consisting of 38,970 bp with a GC content of 42.2%, which corresponded to that (42.1–44.4%) observed for *Parageobacillus* spp. [1]. Similar to other dsDNA bacteriophages, the genome of NIIg3.2 was close-packed—94.6% of the genome was coded for proteins. The analysis of the genome sequence revealed that NIIg3.2 contained 71 probable protein-encoding genes and no genes for tRNA. Notably, an apparent asymmetry in the distribution of the genes on the two DNA strands was observed. With the exception of ORF24-ORF26, encoding proteins potentially related to the lysogenic module of NIIg3.2, all other ORFs were predicted to be transcribed from the same DNA strand (Figure 2).

A bioinformatics analysis revealed that only ORF31 of NIIg3.2 encoded unique protein that had no reliable identity (E-values > 0.001) in the database entries. For NIIg3.2 ORFs that encoded proteins with matches to those in other sequenced viral genomes, the percentage of amino acid identity ranged from 25% to 98%, and, in most cases (45 out of 56 NIIg3.2 ORFs), from 31% to 80% (Appendix A). The vast majority (42 out of 49) of these NIIg3.2 gene products showed the highest similarity to proteins from phages that infect *Bacillus*-group bacteria from the genera *Geobacillus* (32), *Bacillus* (11), *Paenibacillus* (2), *Virgibacillus* (2), *Anoxybacillus* (1), and *Brevibacillus* (1). In addition, four, two, and one NIIg3.2 gene products demonstrated the highest similarity to proteins from *Thermus*, *Clostridium*, and *Staphylococcus* phages, respectively. Furthermore, 14 NIIg3.2 gene products showed reliable identity only to proteins found in *Bacillus*-group bacteria from the genera *Geobacillus* (8), *Parageobacillus* (4), *Anoxybacillus* (1), and *Halobacillus* (1). Based on their homology to biologically defined proteins, 29 ORFs of NIIg3.2 were given a putative functional annotation (Appendix A, Figure 2). As observed in other siphoviruses, the NIIg3.2 genome appeared to have a modular organization, with genes for DNA packaging, structure/morphogenesis, phage–host interactions, lysis/lysogeny, and DNA replication/repair clustered together (Figure 2).

#### 2.2.1. Structural Proteins and Proteomic Analysis

As seen in Figure 2, all NIIg3.2 structural protein-encoding genes were found within a genome cluster (~14-kb) located just downstream of the packaging module. A bioinformatics analysis of the genome sequence of bacteriophage NIIg3.2 allowed the identification of eleven structural genes, including those coding for the head (ORF04–ORF06, ORF08, ORF09), tail (ORF10–ORF12, ORF15, ORF16), and tail fiber (ORF17) proteins (Appendix A). The major capsid protein and major tail protein are two of the main building components constituting the virions of siphoviruses [29]. The major capsid protein (gp06) of NIIg3.2 exhibited the highest similarity to major capsid proteins from *Paenibacillus* phages Dragolir and Wanderer, respectively (Appendix A, Figure 3). The HHpred analysis revealed that NIIg3.2 gp06 best matched the structure of the major capsid protein of Enterobacteria phage HK97 (3QPR_G; probability 100%; E-value 3 × 10^−33^). Three other NIIg3.2 head-related proteins, namely prohead protease (gp05), head–tail connector protein (gp08), and the head closure protein (gp09), showed similarity to the prohead core protein protease of Enterobacteria phage T4 (5JBL_C; probability 96.17%, E-value, 0.65), SPP1 gp15 (7Z4W_e; probability 99.53%, E-value 4 × 10^−13^), and SPP1 gp16 (7Z4W_5; probability 99.86%, E-value 1.2 × 10^−19^), respectively.

The major tail protein (gp12) of NIIg3.2 possessed the maj_tail_phi13 (TIGR01603) conserved domain and had the best HHpred hit to the major tail protein of bacteriophage sp. (6XGR_L; probability, 93.49%; E-value, 5.5). The other NIIg3.2 tail-related proteins included two putative tail components (gp10 and gp11), tape measure protein (gp15), and tail hub protein (gp16). The NIIg3.2 gp10 exhibited phge_HK97_gp10 (TIGR01725) conserved domain. The gp16 contained the Sipho_tail (pfam05709) conserved domain and it had the best HHpred hit to the tail component of *Lactobacillus* phage J-1 (5LY8_A; probability 100%; E-value 1.8 × 10^−30^). No conserved domains via BLASTp analysis were detected in gp11 of NIIg3.2; meanwhile, gp15 possessed the N-terminal tape_meas_TP901 (TIGR01760) and C-terminal SLT (COG3953) conserved domains. As mentioned above, although no tail fibers were clearly visible by TEM (Figure 1), one NIIg3.2 gene coding for the potential tail fiber protein (gp17) was identified by bioinformatics approaches. The gp17 contained the N-terminal Prophage_tail (pfam06605) conserved domain and it had the best HHpred hit to the tail-associated lysin of *Staphylococcus* phage 80alpha (6V8I_CE; probability 99.88%; E-value 3.4 × 10^−20^). 

FASP followed by LC–MS/MS confirmed five NIIg3.2 structural proteins identified by comparative genomics and/or HMM profile comparisons (Appendix A). It was demonstrated that gp05 (prohead protease), gp06 (major capsid protein), gp15 (tape measure protein), gp16 (tail hube protein), and gp17 (tail fiber protein) were present in the virion of NIIg3.2. The indetermination of potential structural proteins, which were identified by bioinformatics approaches (Appendix A) but not detected by proteomics analysis, may be due to the discrepancy of these proteins with sample preparation procedures or/and their low quantity in virions.

#### 2.2.2. Packaging 

The packaging machine of tailed bacteriophages usually displays two essential components: a terminase complex and a portal ring [30]. Most characterized terminases consist of a small subunit (TerS) involved in DNA recognition and a large terminase subunit (TerL) containing the ATPase and the endonuclease activities [31]. The proteins associated with DNA packaging of isolated thermophilic bacteriophage NIIg3.2 included all three aforementioned proteins: the TerS, TerL, and portal protein, which were encoded by ORF01, ORF02, and ORF04, respectively. The NIIg3.2 TerS containing Terminase_4 (pfam05119) and the TerL containing Terminase_1 (pfam03354) conserved domains had the best HHpred hits to TerS (6Z6E_A; probability 99.68%; E-value 4.4 × 10^−16^) and TerL (6Z6D_A; probability 100%; E-value 1.6 × 10^−45^) of Enterobacteria phage HK97, respectively. The portal protein of NIIg3.2 contained the Phage_portal (pfam04860) conserved domain and shared the highest HHpred similarity to the portal protein of *Thermus* phage P7426 (5NGD_B; probability 100%; E-value 2.8 × 10^−33^). 

#### 2.2.3. DNA Replication, Recombination and Repair

A bioinformatics analysis revealed a set of NIIg3.2 genes associated with DNA replication, recombination, and repair (DNA RRR) that were clustered between 25.8 kb and 31.5 kb (Figure 2). The DNA RRR of phage NIIg3.2 included those coding for a Mu Gam-like end protection protein, DNA replication initiation protein (dnaD), DNA replication protein (DnaC), and a Holliday junction resolvase encoded by ORF39, ORF42, ORF43, and ORF52, accordingly. Mu Gam facilitates Mu replication and nonspecifically binds to DNA, protecting linear Mu phage DNA from digestion with nuclease [32]. The Mu Gam-like end protection protein of NIIg3.2 contained conserved Sipho_Gp157 (pfam05565) domain and shared the best HHpred hit with the host-nuclease inhibitor protein Gam from *Desulfovibrio vulgaris* (2P2U_A; probability 99.41%; E-value 6.5 × 10^−12^). The DnaD and DnaC of NIIg3.2 demonstrated the HHpred identity to the DnaD protein from *Bacillus subtilis* str. 168 (8OJJ_A; probability 99.14%; E-value 3.8 × 10^−9^) and DnaC protein from *Escherichia coli* (6QEM_I; probability 99.87%; E-value 1 × 10^−19^), respectively. The best HHpred hit of Holliday junction resolvase, an endonuclease that specifically cleaves Holliday junctions [33], of NIIg3.2 was archaeal Holliday junction resolvase from *Thermus thermophilus* phage 15–6 (7BGS_B; probability 99.59%; E-value 2 × 10^−42^). Additionally, HNH endonuclease, a site-specific DNA endonuclease promoting the lateral transfer of its own coding region and flanking DNA between genomes by a recombination-dependent process termed homing [34], was encoded by ORF71 in the genome of phage NIIg3.2. The gp71 exhibited a HNH (pfam01844) conserved domain, and with a probability of 99.84% and E-value of 4.5 × 10^−19^, showed the highest HHpred identity to the HNH endonuclease (5H0M_A) from *Geobacillus* virus E2.

#### 2.2.4. Transcription, Translation and Nucleotide Metabolism

Based on the amino acid sequence similarity, a number of genes encoding proteins potentially involved in transcription, translation and nucleotide metabolism were present in the genome of phage NIIg3.2. The ORF26 encoded transcriptional repressor containing N-terminal HTH_XRE (cd00093) and C-terminal S24_LexA-like (cd06529) conserved domains, its closest BLASTp viral homologue was transcriptional repressor (YP_001285833.1) from *Geobacillus* virus E2 (64.73% aa identity; E-value, 3 × 10^−94^). The gp27 possessing HTH_XRE (cd00093) and gp32 exhibiting phage_pRha (TIGR02681) conserved domains were transcriptional regulators, which belonged to the families XRE and Rha, respectively. The ORF64 encoded transcriptional activator containing phage_arpU (TIGR01637) conserved domain. In addition, the putative regulatory protein possessing C-terminal HTH_59 (pfam20038) conserved domain was encoded by ORF66. Interestingly, in contrast to other NII3.2 transcriptional regulators, the gp66 had only one BLASTp viral homologue with reliable identity (E-values > 0.001), which was excisionase (YP_001285865) from *Geobacillus* virus E2 (query coverage 45%; identity 72.06%; E-value, 3 × 10^−24^).

One gene associated with nucleotide metabolism was detected in the genome of NIIg3.2. The ORF47 encoded dUTP diphosphatase containing dUTPase_2 (pfam08761) conserved domain, and it shared the best HHpred hit to the dUTPase from *Staphylococcus aureus* (5MYF_D; probability, 99.92%; E-value, 5.2 × 10^−24^). 

#### 2.2.5. Lysis Cassette

Generally, all dsDNA phages accomplish host lysis using a muralytic enzyme (known as an endolysin) and a holin, a small membrane protein that creates pores in the inner or cytoplasmic membrane at a programmed time [35]. The NIIg3.2 lysis cassette consisted of two holins (gp20 and gp21) containing XhlA (pfam10779) and Holin_SPP1 (pfam04688) conserved domains, respectively, and endolysin (N-acetylmuramoyl-L-alanine amidase, gp22) containing N-terminal MurNAc-LAA (cd02696) and C-terminal SPOR (pfam05036) conserved domains. All three lysis proteins were encoded in the canonical order (Figure 2). The presence of two holins, including XhlA domain-containing proteins, has been described in a number of *Bacillus*-group bacteria infecting phages [24,25,36,37]. It is likely that activation of more than one holin may facilitate the lysis of cells grown in different conditions or different phage hosts [36].

#### 2.2.6. Lysogeny and Auxiliary Metabolic Genes

The lysogeny module of NIIg3.2 was found downstream of the lysis cassette. Unsurprisingly, since lysogeny-related genes usually are in the opposite orientation on the complementary strand relative to the other functional modules [38], the lysogeny-related gene of NIIg3.2, which encoded integrase (gp25), was found on the complementary strand (Figure 2). Phage integrases are site specific recombinases that mediate recombination between the phage genome and the bacterial chromosome [39]. The NIIg3.2 integrase containing N-terminal Ser_Recombinase (cd00338) and C-terminal Smc (COG1196) conserved domains was predicted to belong to the integrase family of serine recombinases [40]. It is also likely that transcriptional repressor (gp26), which was encoded on the complementary strand immediately downstream the integrase gene, could play a crucial role in transcription regulation of NIIg3.2 genes, including integrase-encoding *g25*. 

In addition, bioinformatics analysis revealed that NIIg3.2 ORF50 encoded potential auxiliary metabolic genes (AGMs) related to pathogenicity—YopX family protein containing YopX (pfam09643) conserved domain. The HHpred analysis demonstrated gp50 similarity to the YopX protein from *Bacillus subtilis* (2I2L_B; probability, 99.92%; E-value, 1 × 10^−24^) meanwhile the closest BLASTp homologue of gp50 was hypothetical protein (YP_009223844.1) from *Geobacillus* virus E3 (query coverage 98%; identity 62.02%; E-value, 2 × 10^−45^).

### 2.3. Phylogenetic Analysis

To elucidate the phylogenetic relationship between bacteriophage NIIg3.2 and its closest relatives, we conducted a comparative analysis of specific protein-encoding genes commonly employed for the analysis of the evolutionary relationships among bacterial viruses [41]. Phylogenetic trees were constructed based on the alignment of the NIIg3.2 major capsid protein, terminase large subunit, tape measure protein, portal protein, and their respective amino acid sequences with those identified through BLASTp homology searches (Figure 3). The resulting phylogenetic trees consistently demonstrated that NIIg3.2 is distantly related to any previously sequenced bacteriophages, forming a distinct branch in the neighbor-joining phylogenetic trees. As seen in Figure 3, NIIg3.2 appears to occupy an intermediate position between siphophages belonging to the genera *Wbetavirus*, *Dragolirvirus*, *Wanderervirus* and unclassified siphoviruses within the class *Caudoviricetes*.

To gain a comprehensive understanding of the phylogenetic relationships of NIIg3.2 and its closest relatives, a comparative total proteome comparison was performed using the ViPTree web service. It should be noted that the virus–host database utilized by the ViPTree lacked the genome sequences of several phages sharing phylogenetic relatedness with NIIg3.2; therefore, those were added into the query alongside the genome of NIIg3.2. Based on the results of a whole-proteome alignment, our findings demonstrated that phage NIIg3.2 exhibited the closest relationship to unclassified *Geobacillus* viruses, namely vB_GthS_PK5.2 (OP341629), GVE2 (NC_009552), vB_GthS_PT9.1 (OP341630), vB_GthS_NIIg9.7 (OP341624), vB_GthS_PK5.1 (OP341628), vB_GthS_PK3.5 (OP341626), and vB_GthS_PK3.6 (OP341627) (Figure 4). Nevertheless, the phylogenetic relationships between NIIg3.2 and the aforementioned *Geobacillus* bacteriophages were distant, indicating that NIIg3.2 represents an evolutionarily distant lineage within the siphovirus group.

To determine the most homologous regions in the genomes of NIIg3.2 and its closest relatives, namely the *Geobacillus* phages vB_GthS_PK5.2 and GVE2, the genome alignment was performed using ViPTree. Genomes of the bacteriophages shared several regions of nucleotide similarity that cover the essential structural and virion morphogenesis protein-encoding genes, as well as genes related to lysis, lysogeny, and DNA metabolism (Figure 5). Nevertheless, the nucleotide-based virus overall nucleotide sequence identity between NIIg3.2 and its closest relatives was quite low and ranged from 15.5% (NIIg3.2 vs. vB_GthS_PK5.1) to 9.6% (NIIg3.2 vs. vB_GthS_PK3.6) (Appendix A).

According to the Bacterial and Archaeal Viruses Subcommittee (BAVS) of the International Committee on Taxonomy of Viruses (ICTV), two phages could be assigned to the same species if their genomes are more than 95% identical, while a genus is described as a cohesive group of viruses containing a high degree (>70%) of nucleotide identity of the full genome length [42]. Following this (and based on the results of the comparative genome sequence analysis performed during this study), we consider that bacteriophage NIIg3.2 is a singleton virus to date and cannot be classified to any genus currently recognized by ICTV, and likely represents a new one within the siphoviruses.

## 3. Discussion

Endolysins encoded by thermophilic bacteriophages have a special attention because of their high (thermo)stability and unusually broad lytic activity against Gram-negative and Gram-positive bacteria [16]. Comparative analysis of endolysin (N-acetylmuramoyl-L-alanine amidase) from *Parageobacillus* bacteriophage NIIg3.2 revealed that its closest BLASTp viral homologue (80.69% aa identity; E-value, 1 × 10^−134^) was well characterized endolysin (YP_001285830.1) from *Geobacilus* virus E2. It was demonstrated that E2 recombinant endolysin was thermostable, exhibited good tolerances at acid pH values, and played important roles in the lysis process of the host cells [17,18]. Thus, the endolysin from the phage NIIg3.2 could also be an attractive tool in applications in molecular biology as well as in industry applications, but further investigation is required to confirm these speculations.

On the other hand, bacteriophages also could be extensively used as effective antimicrobial agents against detrimental bacteria. However, such bacteriophages must be virulent (strictly lytic), thus lacking the ability to lysogenize targeted hosts and transfer genes associated with resistance and virulence [13,43]. Bioinformatics analysis revealed that ORF50 of *Parageobacillus* phage NIIg3.2 encoded YopX family protein. The *Yersinia* outer protein X (YopX) is a protein of plasmid origin found in bacteria, and it is known as a potential virulence factor against eukaryotes [44]. Additionally, YopX-like proteins, which have been reported as class II auxiliary metabolic genes (AMGs) are presented in the replication module of a variety of phages [22,45,46], and are potentially involved in the enhancement of host functionality to improve viral propagation [47]. However, whether gp50 of NIIg3.2 affects viral propagation and/or its host pathogenicity, remains unclear.

The life cycle of phage NIIg3.2 remains an open question. Two lysogeny-related genes of NIIg3.2, encoding integrase (ORF25) and transcriptional repressor (ORF26), were presented on the complementary strand of DNA of phage NIIg3.2 (Figure 2). The gp26 shared the best HHpred hit to the lambda repressor (3BDN_B) from Enterobacteria phage lambda (probability, 99.85%; E-value, 4.7 × 10^−19^). The binding of lambda repressor to operator regions O_R_ and O_L_ on the phage chromosome is required during lysogenic growth, when the integrated prophage replicates as part of the host chromosome [48]. The integrase (gp25) of NIIg3.2, which showed the highest BLASTp homology exceptionally to integrases from *Bacillus*-group bacteria infecting phages (Appendix A), is predicted to belong to site-specific serine recombinases (also known as the resolvase family), which cut all DNA strands in advance of strand exchange and religation [49].

However, despite the presence of lysogeny-related genes in the genome of NIIg3.2, the phage host range and temperature range determination experiments suggested little about the lytic cycle of this phage. To determine whether lysogens could be recovered from NIIg3.2 infections, cells from a spot where NIIg3.2 particles had infected a lawn of *P. toebii* strain NIIg-3 were recovered and grown on solid media. Bacterial growth was observed, and two independent colonies were restreaked twice more and then patched onto *P. toebii* strain NIIg-3 lawns to test for phage release. None of the colonies recovered showed phage release (unreported data). Thus, although lysogeny-related genes, which are typically observed in temperate phages [50], were present in the genome of NIIg3.2, there is no evidence that this phage is capable of lysogenizing *P. toebii* strain NIIg-3.

Furthermore, although a number of genes encoding proteins associated with DNA replication, recombination, and repair (DNA RRR) were identified in the genome of NIIg3.2, including Mu Gam-like end protection protein (ORF39), dnaD (ORF42), dnaC (ORF43), Holliday junction resolvase (ORF52), and HNH endonuclease (ORF71), no homologues to essential DNA RRR-associated proteins (for example, single-stranded DNA binding (SSB) protein, helicase, and DNA polymerase) were detected using bioinformatics approaches. The absence of aforementioned essential genes in the genome of NIIg3.2 suggests that either these genes are highly divergent from their homologues, or this phage most likely uses a number of DNA RRR proteins, including DNA polymerase, of the host cell, as it was observed in a number of phages including *Bacillus* phage SPP1 [51].

Interestingly, the results of the host range determination experiments demonstrated that NIIg3.2 infected *P. toebii* strain NIIg-3 and four strains of *G. thermodenitrificans*, but it was not active against all strains of *P. caldoxylosilyticus* and *P. thermoglucosidasius* tested. Although the ability of the phage to infect bacteria from phylogenetically closely related genera has been demonstrated [52], what factors determine phage specificity, especially in the case of thermophilic bacteriophages, is still an open question. It is known that viral host range could be determined by a number of molecular mechanisms [53], and it is likely that the initial stage of phage infection could play an essential role. The results of the bioinformatics analysis suggested that one of the most important NIIg3.2 proteins involved in the initial steps of host recognition and adsorption could be tail fiber protein (gp17).

The phylogentic analysis (Appendix A) demonstrated that gp17 of NIIg3.2 was the most closely related to tail fiber proteins from thermophilic *Geobacillus* bacteriophages vB_GthS_PK3.5 (PK3.5), vB_GthS_PK3.6 (PK3.6), vB_GthS_PT9.1 (PT9.1), vB_GthS_NIIg9.7 (NIIg9.7), and vB_GthS_PK5.2 (PK5.2) which were isolated from the same compost heaps as NIIg3.2 [25], (unpublished data). However, the identity of NIIg3.2 gp17 in the amino acids level to tail fiber proteins of aforementioned bacteriophages was quite low (Appendix A). The analysis of the conserved domains of NIIg3.2 closest homologues demonstrated that all tail fiber proteins contained the N-terminal Prophage_tail (pfam06605) conserved domain. However, with the exception of proteins from NIIg9.7 and PT9.1, the architecture of the central part and the C-terminus of tail fiber proteins were different (Appendix A). Taking into account that phages NIIg3.2, PK3.5, PK3.6, PT9.1, NIIg9.7 and PK5.2 (phage infects only *G. thermodenitrificans* strain PK-5, unpublished data) infect different strains of *Geobacillus* and/or *Parageobacillus*, it is likely that different profiles of the phage hosts may be determined by differences in the amino acid sequences of the tail fiber proteins and/or by other viral proteins potentially involved in phage–host interaction or even later stages of the phage replication cycle. Further studies would be needed to gain a deeper understanding of the interactions between NIIg3.2 phage and its hosts.

## 4. Materials and Methods

### 4.1. Phages and Bacterial Strains

Bacteriophage NIIg3.2 was originally isolated from soil samples collected from compost heaps at Vilnius University Botanical Garden, Vingis Park, Vilnius, Lithuania (54.682912, 25.232532). *P. toebii* strain NIIg-3 was used as the host for NIIg3.2 isolation, propagation and phage growth experiments. The bacterial strains used in this study for host range determination of NIIg-3 are listed in Appendix A. For all of the phage experiments, the bacteria were cultivated in Luria–Bertani (LB) broth agar (Formedium) and/or gellan (PanReac Applichem). Solid plates were prepared by adding the appropriate amount of agar (Formedium) or gellan (PanReac Applichem) to the liquid medium. Phage host range determination experiments were performed in triplicate to confirm the results.

### 4.2. Phage Isolation, Propagation and Purification Techniques

Phage isolation was performed by using the enrichment of phages in the source material technique as described previously [25]. Phage titration and efficiency of plating experiments were performed by using the soft agar overlay method with minor modifications described by Šimoliūnas et al. [25]. Phage titration and efficiency of plating experiments were performed in triplicate to confirm the results. Bacteriophage was purified by performing five consecutive transfers of phages from individual plaques to new bacterial cell lawns. Notably, as isolated thermophilic bacterial strains were growing poorly in a liquid broth, the propagation of bacteriophage was performed by the soft agar overlay method with minor modifications described by Šimoliūnas et al. [25]. Further phage purification was performed using a cesium chloride (CsCl) density gradient centrifugation, as described previously [54].

### 4.3. Transmission Electron Microscopy

The CsCl density gradient-purified phage particles were diluted to approximately 10^11^ PFU/mL with distilled water, and 10 µL of the sample was directly applied on the carbon-coated nickel grid (Agar Scientific, Essex, UK). After 1 min, the excess liquid was drained with filter paper and stained with two successive drops of 2% uranyl acetate (pH 4.5) for 1 min. The sample was then dried and examined using a Tecnai G2 F20 X-TWIN transmission electron microscope (FEI, Hillsboro, OR, USA). The magnification value was valid for TEM monitor only, as it was depended on the screen size, zooming, or printed micrograph dimensions.

### 4.4. DNA Isolation

The aliquots of high-titer (~10^11^ PFU/mL) phage suspension were subjected to phenol/chloroform extraction and ethanol precipitation, as described by Carlson and Miller [55] with minor modifications. Briefly, the DNaseI and RNase A Treatment was used before phenol/chloroform extraction (optional): DNaseI (1U/µL) was combined with 1 µg of DNA. The mixture was incubated at 37 °C for 30 min. DNaseI was subsequently inactivated by the addition of 50 mM EDTA, followed by incubation at 65 °C for 10 min. RNase A (10 mg/mL) was utilized at concentrations ranging from 1 µg/mL to 100 µg/mL. The mixture was incubated at 37 °C for 30 min. Additionally, deproteinization with Proteinase K was used before phenol-chloroform extraction: 10 µL of 10% SDS and 1 µL of Proteinase K (20 mg/mL) were added to 200 µL of phage suspension. The mixture was thoroughly mixed and incubated at 55 °C for 1 h. Afterwards, DNA phenol was mixed with an equal volume of phage sample, followed by thorough hand shaking or vortexing. Subsequently, the mixture was centrifuged at room temperature for 5 min at 16,000× *g*. The upper aqueous phase was carefully removed and transferred to a fresh tube. An equal volume of phenol-chloroform-isoamyl alcohol (25:24:1) was added, followed by thorough shaking. The mixture was then centrifuged at room temperature for 5 min at 16,000× *g*. This step was repeated a minimum of two times. The upper aqueous phase was once again carefully removed and transferred to a fresh tube. An equal volume of chloroform–isoamyl alcohol (24:1) was added, followed by thorough shaking and subsequent centrifugation at room temperature for 5 min at 16,000× *g*. This step was also repeated a minimum of two times. Genomic DNA was precipitated by adding 1/10 of the volume of 5 M NaCl and 2.5 times the volume of 96% ethanol. The mixture was then incubated at −20 °C for a minimum of 2 h (preferably overnight for 16–18 h). Subsequently, the samples were centrifuged at 4 °C for 20 min at 16,000× *g* to pellet the genomic DNA. Following centrifugation, the supernatant was carefully removed without disturbing the DNA pellet. The DNA pellets were dried in a SpeedVac concentrator for 2–5 min or at room temperature for 20–30 min. Finally, the DNA pellets were resuspended in 100–300 µL of nuclease-free water by pipetting. The isolated phage DNA was subsequently used for PCR, or it was subjected to genome sequencing.

### 4.5. Genome Sequencing and Analysis

The complete genome sequence of bacteriophage NIIg3.2 was determined using Illumina DNA sequencing technology at Macrogen (Amsterdam, The Netherlands). DNA libraries were prepared using TruSeq DNA PCR Free (350) library preparation. The 151 bp length paired-end sequence reads were generated using the NovaSeq (6000) platform. 

FASTQ read sequence files were generated using bcl2fastq version 2.20 (Illumina). The quality of the raw reads was evaluated using FASTQC quality control tool version 0.11.9 [56] (available online: http://www.bioinformatics.babraham.ac.uk/projects/fastqc/, accessed on 11 July 2023). Low quality bases and adapters were trimmed using TrimGalore version 0.6.6 [57] (available online: https://www.bioinformatics.babraham.ac.uk/projects/trim_galore/, accessed on 11 July 2023) using standard parameters. Samples were downsampled using reformat.sh tool from BBMAP package version 38.96 (available online: https://sourceforge.net/projects/bbmap/files/, accessed on 11 July 2023) up to approx. 70M reads each. The quality of the reads was improved using BayesHammer [58] bundled with the SPAdes package and genomes were assembled using SPAdes packages version 3.13.1 [59]. BBMAP package (v38.96) was used to evaluate mapping rate and coverage. The reads of bacteriophage were assembled into a single linear contig of 39,047 bp (97.527 overall mapping rate; 536.754 average coverage). The ends of the assembled contig were confirmed using PCR, followed by Sanger sequencing reactions at Macrogen (Amsterdam, The Netherlands). A PCR fragment was obtained by the amplification of NIIg3.2 phage wild-type DNA using NIIg3.2_F1 5′-GGCGATATCGCAGCTGGTGCTC-3′ and NIIg3.2_R1 5′-GCTTGATCTTCTGTACTGACGCG-3′ primers. PhageTerm [60] was used for determination of phage termini. No defined genomic termini were identified, and to preserve gene contiguity, the genome start point was selected from the predicted terminase small subunit gene.

The open reading frames (ORFs) were predicted with Geneious Prime version 2023.1 (available online: http://www.geneious.com/, accessed on 11 July 2023) using a minimum ORF size of 60 nt. The analysis of the genome sequences was performed using BLASTp, BLASTx, Fasta-Protein, Fasta-Nucleotide, Transeq (available online: http://www.ebi.ac.uk/Tools/st/emboss_transeq/, accessed on 11 July 2023), Clustal Omega (available online: http://www.ebi.ac.uk/Tools/msa/clustalo/, accessed on 11 July 2023), and DNA sequence editor available online: http://www.biocourseware.com/iphone/dnaseqeditor/index.htm/, accessed on 11 July 2023), as well as HHPred and HHblits, [61,62]. The tRNAscan-SE 2.0 (available online: http://lowelab.ucsc.edu/tRNAscan-SE/, accessed on 11 July 2023) and ARAGORN (available online: http://www.ansikte.se/ARAGORN/, accessed on 11 July 2023) were used to search for tRNAs. Neighbor-joining phylogenetic tree analysis was conducted using MEGA version 5 [63]. ViPTree [64] version 3.6 was used for the total proteome comparisons (available online: https://www.genome.jp/viptree/, accessed on 15 June 2023). The overall nucleotide sequence identity was calculated using intergenomic distance calculator—VIRIDIC [65].

### 4.6. Analysis of Structural Proteins

An analysis of the structural proteins of phage virions was performed using a modified filter-aided sample preparation (FASP) protocol, followed by LC-MS/MS analysis, as described previously [66].

### 4.7. Nucleotide Sequence Accession Numbers

The complete genome sequence of *Parageobacillus* bacteriophage vB_GthS_NIIg3.2 was deposited in the EMBL nucleotide sequence database under accession number OP341623.

## 5. Conclusions

In conclusion, it was shown that bacteriophage NIIg3.2 is a thermophilic siphovirus-infecting bacteria of the genera *Geobacillus* and *Parageobacillus*. Additionally, it possess no close genetic homology to previously described phages and potentially represents a new genus within the class *Caudoviricetes*. Thus, the results of this study may provide new insights that deepen our understanding of *Bacillus*-group phage genetics and phage–host interactions in dynamic ecosystems, such as compost heaps.

## Figures and Tables

**Figure 1 ijms-24-13980-f001:**
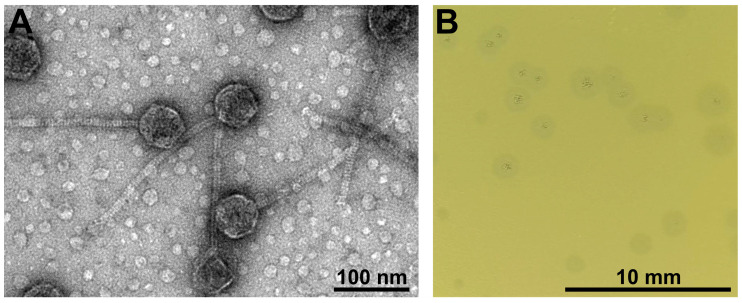
Electron micrographs of NIIg3.2 virions (**A**) and morphology of plaques formed by phage NIIg3.2 (**B**). The TEM ×35,000. The morphology of the plaque forming units was monitored after 16 h of incubation at 55 °C on a lawn of *Parageobacillus toebii* strain NIIg-3.

**Figure 2 ijms-24-13980-f002:**
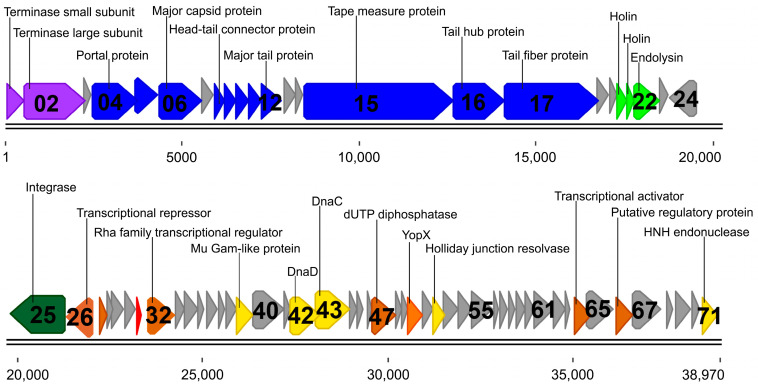
Functional genome map of *Parageobacillus* bacteriophage NIIg3.2. The coding capacity of the genome is shown. Numbers indicate ORF position in genome, functions are assigned according to the characterized ORFs in NCBI database and HHpred analysis. The color code is as follows: yellow—DNA replication, recombination, and repair; blue—structural proteins, phage–host interactions; purple—DNA packaging; brown—transcription, translation, nucleotide metabolism; light green—lysis; dark green—lysogeny; orange—auxiliary metabolic genes; grey—conserved hypothetical proteins; red— hypothetical proteins with no reliable identity when compared to database entries. Annotations for unnamed ORFs presented in Figure 2 are presented in Appendix A.

**Figure 3 ijms-24-13980-f003:**
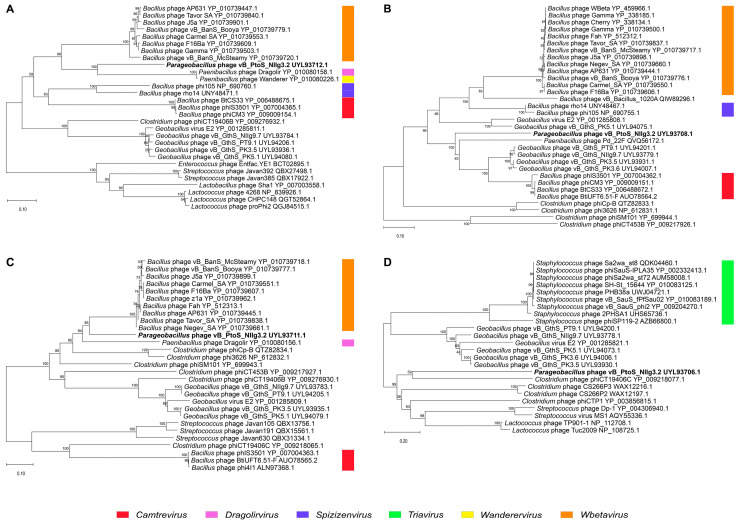
Neighbor-joining tree analysis based on the alignment of the amino acid sequences of *Parageobacillus* bacteriophage NIIg3.2: (**A**) major capsid protein, (**B**) terminase large subunit, (**C**) portal protein and (**D**) tape measure protein (TMP). The phylogenetic analysis was conducted using MEGA version 5. The percentage of replicate trees in which the associated taxa clustered together in the bootstrap test is shown next to the branches. Bacteriophages without indicated genera are unclassified viruses within the class *Caudoviricetes*.

**Figure 4 ijms-24-13980-f004:**
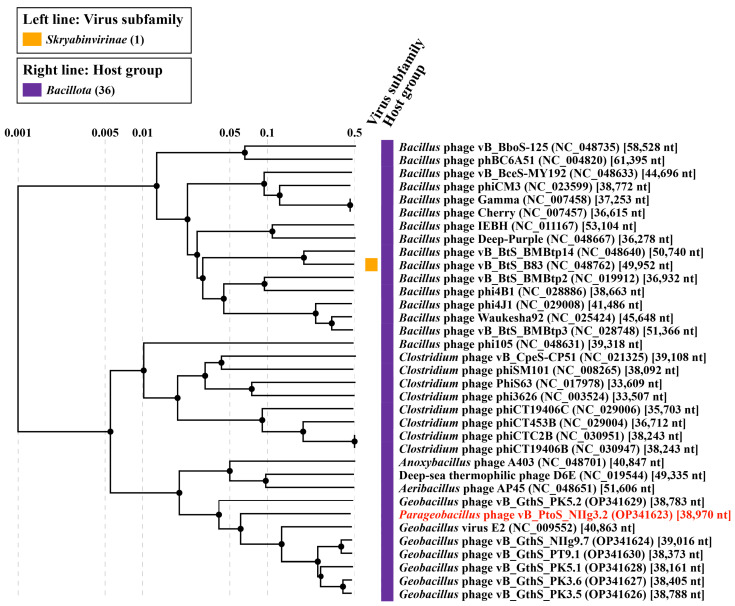
ViPTree generated proteomic tree of *Parageobacillus* phage NIIg3.2 and dsDNA viruses represented in the rectangular view. The tree is constructed by BIONJ based on genomic distance matrixes, and mid-point rooted. Branch lengths are logarithmically scaled from the root of the entire proteomic tree. The numbers at the top represent the log-scaled branch lengths based on the SG (normalized tBLASTx scores) values.

**Figure 5 ijms-24-13980-f005:**
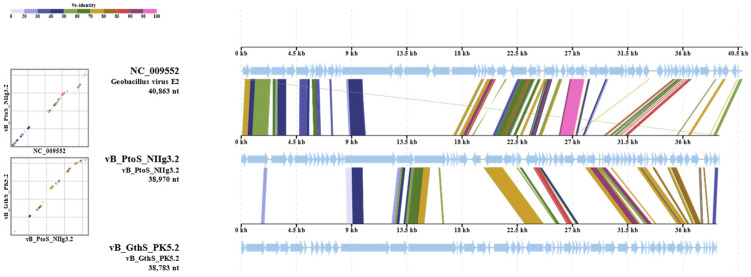
ViPTree generated whole-proteome alignment of *Parageobacillus* phage NIIg3.2 and its closest relatives: *Geobacillus* viruses E2 and vB_GthS_PK5.2. Colored lines in the alignment indicate tBLASTx results (E-value < 0.01). Positions of each sequence are automatically adjusted (i.e., circularly permuted and reverse stranded) for clear representation of collinearity between genomes.

## Data Availability

The complete genome sequence of *Parageobacillus* bacteriophage vB_GthS_NIIg3.2 is available in the EMBL nucleotide sequence database under accession number OP341623.

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
