# Peer review of "Characterization of Parageobacillus Bacteriophage vB_PtoS_NIIg3.2—A Representative of a New Genus within Thermophilic Siphoviruses"

_ijms, 2023, doi:10.3390/ijms241813980_

Round 1
Reviewer 1 Report
Title: Parageobacillus bacteriophage vB_PtoS_NIIg3.2 – a representative of a new genus within thermophilic siphoviruses
This manuscript is not ready to publish in the journal as many weak points were presented in it. However, I do believe that if they can improve the manuscripts following all comments. It might have a chance to publish in the journal.
Comments
1. Topic: The title is not clear and should be modified.
2. Line 29-30: Please re-write or modify the sentence.
3. Line 30-33: Please re-write or modify the sentence.
4. Please add the conclusion of the study at the end of the abstract.
5. Line 34: Keywords: Please add “NIIg3.2” as one of the keywords.
6. Line 37: “Gram-strain-positive” should be replaced by Gram-positive bacteria
7. Please add some information of “NIIg3.2” such as the origin of the phage (isolation) in the introduction
8. Please identify the objective of this study. It may be better if the author uses passive voice instead of, we present ……
9. Line 84-87: Please re-write or modify the sentence
10. Figure 1: Please add magnification of the TEM images
11. Figure 1: Please use arrows to identify the head or tail of the phages as well as plague.
12. Figure 3: The size of the letters is small. Please increase the size of the letters or modify.
13. The names of genes should be written in italic. Please check.
14. Some sentences in discussion should be removed to introduction. The first and the second paragraphs of the discussion should be summarized to 2-3 sentences.
15. The authors shouldn’t repeat the results in the discussion.
16. Please add the information of the discussion. Try to compare the results (the author’s hypothesis) with other finding by other researchers.
17. Line 443: Phage Isolation: How much concentrations of the bacteria used in the phage isolation?
18. What is CsCl step gradient?
19. Line 453: Transmission Electron Microscopy: How many magnifications of the TEM?
20. Line 460: Please add more information of DNA Isolation. It is beneficial to the reader as well as citation.
21. Please add statistical analysis such as descriptive
22. Please re-write the conclusion. It is not related to the results. Please add more information.
23. There are many references. In general, there are 30-40 references for each manuscript (research articles). Please delete some unnecessary references or old references.
24. The references of 2022,2021, and 2023 are suggested to be cited.

This manuscript is well-written by the authors. By the way, please correct or modify some sentences.
Author Response
Dear Reviewer,
on behalf of all co-authors, I would like to thank you for your efforts to improve our manuscript. We have responded to the comments and have revised the paper in the light of them.
Best wishes,
dr. Eugenijus Šimoliūnas

Reviewer 2 Report
This manuscript reports a new thermostable bacteriophage. It includes an EM, host range study, and a sequence annotation. There is a GenBank file. I focused on the sequence annotation, since that's my specialty. They used HHpred as their primary annotation tool, which conforms to my recommendations for how to do it. I also ran it through my pipeline and found much the same. There are a few nuances by way of the choice of how to annotate I'll note below, but these are matters of judgment and I wouldn't insist that they adopt my recommendation. All in all, I'd say this paper is in the top 10% in terms of thoroughness and accuracy of all the papers sent to me to review of this kind.
There are only two things that appear to be outright errors:
Line 512 quoted the wrong accession number for their phage; it should have been OP341623.
On fig 2. gp47 is colored brown. I think you meant to color it orange.
Other things I noticed that you may or may not want to consider:
You might want to mention that HK97-like capsid protein includes the N-terminal scaffold domain.
You also might look to see if it has the chainmail linking residues. If it's possibly crosslinked, that's worth noting with respect to its temperature stability.
There are more genes annotated in your GenBank file and in your supplement than on your figure. You might note in the legend that annotation for additional genes is found in the supplemental table.
gp20 is annotated as a hemolysin, and in the text it is revised to be a holin. It is actually a stronger HHpred match to the SPP1 gp24.1 holin than to the hemolysin model. I'm concerned that the enthusiasm for finding virulence factors has resulted in relatively broadly defined hemolysin families, which raise lots of false positives. At least my collaborators have sent me preliminary annotations with lots of hemolysins that didn't stand up to closer inspection. Two holins are actually quite common in phages. I'd emphasize the holin annotation on the map and in the GenBank file and supplement.
gp26 is annotated as transcription factor. Look one more hit down the hhpred list and find it matches pdb:3BDN_A lambda cI, including the lexA autoproteolytic domain that renders phages with this immunity repressor inducible by the SOS system. Your discussion mentions lexA and implies that this is the immunity repressor by naming it as part of the lysogeny operon. But you kind of beat around the bush on this winding up by saying it may play a crucial role in regulation of the integrase. That's your immunity repressor. It plays a crucial role in the regulation of everything. Naming it as such would also help focus attention on the intergene region between gp26 and gp27 as the bistable lytic/lysogeny switch, and demarcates the operon starting with gp27 as early rightwards.
Line 93. refers to "several" proteins encoding tail fibers. You have only one tail fiber gene. Here you should just say tail hub and tail fiber encoding genes have been identified. When you get to discussing gp16, instead of the Pfam match, I'd go one down the list and cite the match to the pdb model for lactobacillus phage J-1. There you can look up the pdb entry, follow it to the paper and see in 3D detail that it's basically tail hub (SPP1 dit), with an extra domain inserted that provides an extra cell wall binding contact on the bottom of the tail hub. That would be in addition to the tail fiber that also mounts on dit.
Line 200 DNA replication and repair genes were scattered throughout the genome.
The genes you mention are not really scattered throughout the genome; the are clustered between 25.8 kb and 31.5 kb in the early rightwards operon. What you might more exactly mean is embedded within an extensive array of small hypothetical protein frames in the early rightwards operon. I see by the coloring on the figure you considered the HNH endonuclease to be of this class. I probably wouldn't. As you noted, HNH nucleases are mostly thought of as selfish elements. I don't know of any cases where an HNH nuclease plays a role in DNA replication or repair. I do know of cases where they are involved in packaging, and there it is, right in front of the small terminase. However, HNH nucleases are notoriously scattered everywhere in phage genomes, so making any inference based on their location is hazardous.
I liked the tree work.
Author Response

(The authors gave the same response as above.)
